# REASON TO BEHAVE: ACHIEVING HUMAN-LEVEL TASK EXECUTION FOR PHYSICS-BASED CHARACTERS

## ABSTRACT

Two challenges have consistently been key in human-like agent research: 1) Many jobs require human-level reasoning and introspection capabilities to discern underlying patterns. 2) Controlling complex modeled humanoid characters in indeterministic black-box physical environments demands a powerful controller to exhibit human-like movement and commonsensical behavior. To the end, we introduce "Reason to Behave", a synergistic framework combining large language models (LLMs) based introspective reasoner with an enhanced controller. The reasoner empowers the agent with extensive world knowledge and semantic insights, enhancing contextual interpretation and reasoning formulating a code-based action plan to bridge the gap between high-level instructions and the underlying simulator. The steerable controller embeds motion-phase representation into adversarial motion prior to the precise timing of diverse life-like behaviors, allowing rapid mastery over 100 semantically distinct actions, ranging from locomotion, dance, and sport to challenging specialized maneuvers, preventing mode collapse during skill learning. Without any reward-shaping or training, our character intuitively performs commonsensical behavior, excelling in many real-world tasks from navigation to more complicated challenges like room escaping and pressure plate puzzle. Videos, codes are available at `https://sites.google.com/view/reasontobehave`.

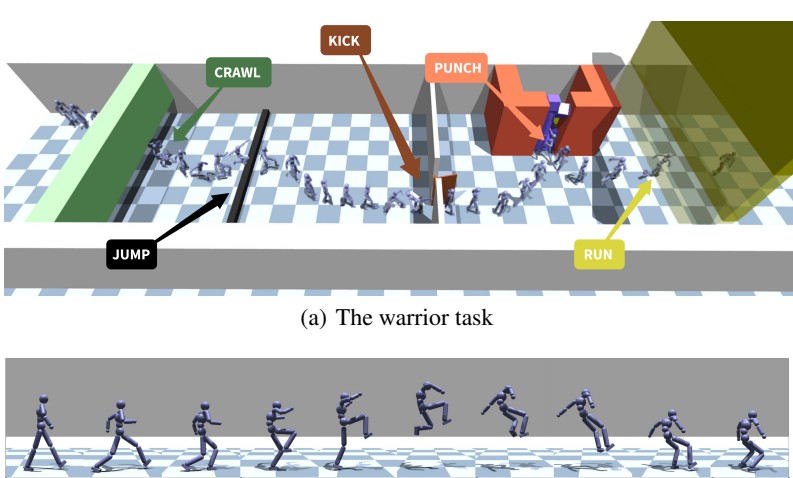

(a) The warrior task

(b) Run then jump skills with its motion phase divide

Figure 1: (a): our character can handle long horizon and reasoning required tasks such as *Warrior*, where the humanoid character has to figure out a correct execution logic and possesses a steerable low-level control to orchestrate different skills. (b): we propose a versatile controller that leverages skill embedding with unified motion phase information to ensure perfect low-execution in a more precise manner.

# 1 INTRODUCTION

Human possess a wide range of motor skills and exhibit unparalleled abilities in reasoning, understanding their environment**?**, and seamlessly interacting with the real world. Developing physically simulated characters that mimic humans Peng et al. (2018) has long been considered an ideal goal for anthropomorphic embodied AI.

Most earlier works employ hierarchical reinforcement learning to endow physically simulated humanoid agents with such capabilitiesPeng et al. (2022)Juravsky et al. (2022), where separated high-level policies are constructed for different tasks and share one directable low-level controller. The low-level controller commonly learned from imitation learning can perform various general-purpose life-like motor skills. This architecture has made strides. The character can tackle tasks like navigation in human-like behavior. However, enabling humanoid characters to finish tasks that require precise reasoning or understanding of object semantics in the environment is still challenging. Although investing significant time in reward engineering may mitigate it, the hard-learned knowledge struggles to be transferred to other tasks. All these stand in stark contrast to human capabilities of versatility in task handling and introspective reasoning due to the absence of foundational world knowledge and an inherent understanding of their abilities and embodiment.

Large language Models (LLMs) trained on massive internet data integrate human-level world knowledge, deeply understand the worldWei et al. (2022), and exhibit logical reasoning abilities. Some researchers leverage as a top-level reasoner in a manner reminiscent of human reasoning, which interprets all contextual information into executable low-level plans, yielding promising task performance in many domains, including robotics, embodied AI gaming, and general task solvingWang et al. (2023). However, the feasibility of transferring such reasoners to a more concrete 3D physically simulated environment Makoviychuk et al. (2021) remains open due to the indeterministic, black-box nature of physics engines and the anthropomorphic nature of humanoids. An imperfect low-level execution could ruin a perfect high-level plan. Commonly, humanoid controllers struggle with accurately carrying out the reasoner's instruction in a non-differentiable physically simulated environment. Also, as a distinctly human-like embodied agent with high-dimensional action space, plain controllers are challenging to manage its large degrees of freedomPeng et al. (2018). This becomes even more complex when requiring the endowment of the capability to perform lifelike, natural movements during task completion.

To overcome the challenges above, we propose an integrated framework, "Reason to Behave", which synergizes an introspective reasoner with a steerable low-level controller to perform a human-like task execution for physics-based characters. Specifically, the reasoner maintains three components, "Planner, Picker, and Programmer", to interpret perception seamlessly and selectively translate high-level reasoning output into precise low-level execution control logic. To ensure the infallibility of low execution, we introduce a phase-enhanced low-level controller by incorporating unified motion phase representation into the skill-conditioned adversarial motion prior to imitation learning Juravsky et al. (2022). The adversarial motion prior enables the humanoid to perform the human-like and natural motor skill in a non-differentiable physically simulated environment. In contrast, the enhancement of conditioned skill embedding and motion phase representation empowers the controller to expose two critical interfaces for more steerable and precise control: 1) specify which skill to execute. 2) determine the start time and repetition count for execution. Furthermore, we verified that combining motion phase information incentivizes policy learning by recognizing the different stages of every reference motion. We have designed sensor APIs and skill APIs that make it easy for the introspective reasoner to invoke the low-level controller. The introspective reasoner can directly control the humanoid agent's interaction with the environment in Python code output. This design achieves alignment from the high-level reasoning of LLMs to the low-level execution of physics-based tasks. Given any task, the system follows a physiological control mechanism. As a collection of information from the current body's state and the environment, the reasoner interprets information and then triggers a low-level controller to execute specific motor skills in a control loop with continuous environmental feedback. The overall contributions of this paper are as follows:

- We present a novel system for achieving human-like task execution for physics-based characters, which synergizes two essential modules: 1) an LLMs-based introspective reasoner with three architecturally organized components for task interpretation and instructions

generation. 2) an enhanced versatile low-level controller to translate instruction into real low-execution.

- We propose a steerable physics-based controller that leverages conditional adversarial motion prior to disparate skill embeddings with unified motion phase representation. This design facilitates the rapid learning of over 100 semantically distinct motor skills by preventing mode collapse and renders the utmost precise control, ensuring a perfect low-level execution.

- We open-source our versatile humanoid character control policy with accessible various control APIs, hoping to facilitate the transfer of powerful reasoning models into the humanoid character in physical simulation environments.

To demonstrate the system's effectiveness, we evaluate our approach to several humanoid tasks, from rudimentary tasks like navigation and knock-over to reasoning-required tasks like pressure plate puzzles and escape games. Our framework demonstrates an exceptionally high success rate in task completion across all tasks. The manipulated humanoid character shows a human-like movement and commonsensical behavior during tasks, highly consistent with human reasoning and execution.

## 2 RELATED WORKS

**Skill learning while task solving**. In some earlier research, the challenge of directing humanoids to accomplish tasks within a physically constrained environment was addressed by computing the equations of motion. These approaches regard the defined task as a trajectory optimization problem, aligning the character's actions towards the optimization objective. However, since the absence of a reference motion, the character may render unnatural behaviors. Imitation learning-based methods ease this problem by assigning certain motion data to instruct the character "How should your movement be like when doing the task", which has become an important technique in the domain of character animation in recent years. Given a reference motion, control policy is expected to control the character generating an almost identical movement while concurrently solving one specific task Peng et al. (2022)Peng et al. (2018)Peng et al. (2021). It is implemented by designing both a motion-tracking reward and a task reward, following the reinforcement learning paradigm to ensure effective policy training. The subsequent approach introduces adversarial motion prior(AMP) Peng et al. (2021) for generating style mimic reward, successfully empowering the control policy to learn unstructured motor skills from a bunch of reference motions and utilize skills to solve a task imultaneously.Creswell et al. (2018) Nonetheless, the learned knowledge cannot be transferred to other tasks, even if the provided reference motions remain consistent with prior tasks, the policy must relearn both motor skills and tasks from scratch.

**Skill learning then task solving.** In pursuit of bestowing reusability upon acquired motor skills, some works are trying to decouple the motor skills learning from the downstream tasks into separated stages. Predominantly, these approaches Tessler et al. (2023) mainly construct a multi-skill latent space, where a foundational controller serves as a decoder, perpetually translating sampled latent into actions. Subsequently, different task-oriented high-level policies reuse the constructed multi-skill latent space as their action space, manipulating the character to accomplish downstream tasks, which is a typical architecture of hierarchical reinforcement learning. Peng et al. (2022) employs an encoder coordinating with AMP to encode the character's multi-step state transitions into a spherically constricted latent space. To obtain a more directable low-level control, PADLJuravsky et al. (2022) and CALM build a semantic latent space wherein each latent represents a distinct skill. Physics-based VAEWon et al. (2022), in their preliminary stages, trains several motion-tracking policies for each reference motion and collects a "state-action" trajectory to compress skills to a versatile stochastic latent space via a conditional VAE, ready for being utilized. Like the task outlined in "skill learning while task solving", many such methodologies have tested on tasks such as location, strike or Boxing Bag to ascertain their efficacy Peng et al. (2022). Nonetheless, these methods are devoid of human prior knowledge, and the character still often exhibit behaviors that don't align with human decision on intricate tasks. When confronted with more convoluted tasks that require reasoning, the substantial cost of reward shaping struggles to endow the model with the requisite logic for correct task execution.

**Planing then task solving.** The CALM framework Tessler et al. (2023) encompasses modules for executing skills and precise navigation, utilizing a pre-designed logical finite automaton as

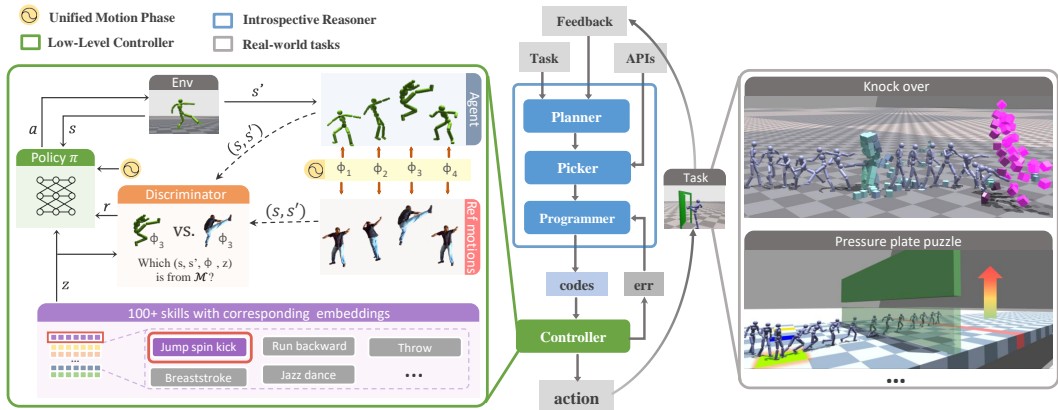

Figure 2: The overview of Reason to Behave framework, which synergizes large language models (LLMs) based introspective reasoner with a low-level controller enhanced by motion phase information.

the pivotal logic control unit to handle downstream tasks. Despite its proficiency in invoking apt skill at opportune moments, this paradigm, which is contingent upon pre-established human-created rules, struggles to achieve generalization across a diverse array of downstream tasks. LLMs? cultivated from an expansive corpus of internet data, are endowed with abundant semantic knowledge, granting it the aptitude to exhibit human-like cognitive reasoning and decision-making capability. DoremiGuo et al. (2023) uses LLMs as a high-level control unit, and assists humanoid tasks of box relocation by continually assessing the box's state to determine the activation of a "replan" event. Yet, in the absence of a potent low-level controller, Doremi is relegated to relying on several crafted scripts of mid-level actions correlated with box transportation, which coordinate three separate policies to render few skills like stand or move. The scarce skills substantially constrict the LLMs' potential to showcase their vast worldly knowledge. In domains outside humanoid research, some works encapsulate LLMs as embodied agentsWei et al. (2022)Wang et al. (2023), substituting humans in open-world games such as Minecraft. Through the integration of an adeptly designed Inner Monologue-based planner, such models can adeptly handle a myriad of in-game tasks, proffering coherent mid-level plans. However, reliance on predefined scripts remains indispensable for each mid-level action. In our approach, we not only architect an LLMs-based reflective planner but also construct a powerful low-level controller congruous with the planner, capable of executing a vast repertoire of semantically distinct skills.

## 3 METHODS

In this section, we will elaborate on modules in our framework according to the following workflow. We first inspect the design details of skill embedding and unified motion phase representation conditioned low-level controller in subsection 3.1, focusing on its . Next, in subsection 3.2 we walk through the introspective reasoner and explain how it gradually corrects its logic with continuous environment feedback. We also briefly describe the self-adjustment stage, an auxiliary procedure to ensure that humanoid interaction skills can successfully engage with the target object in any state, empowered by the introspective reasoner.

### 3.1 LOW-LEVEL CONTROLLER

#### 3.1.1 IMITATION OBJECTIVE BY CONDITIONAL ADVERSARIAL MOTION PRIORS

We propose strong Conditional Adversarial Motion Priors to endow policy with the capability of performing natural and human-like behavior with more precise control. At each timestep, the control policy $\pi(a_t|s_t, z_t, \phi)$ takes three essential inputs: the current states $s_t$ of the whole character body, one skill embedding $z$ to specify which skill to perform, and a unified motion phase representation

$\phi$ to indicate within the current states, the control is expected to execute which stage of the specified skill $z$. To achieve this, we first construct a reference motion dataset that contains a variety of semantically distinct reference motions that represent disparate skills, denoted as $m \in \mathcal{M}$. A motion encoder from a pre-trained MotionCLIP model is applied on $\mathcal{M}$ to generate a discrete skill codebook $\mathcal{Z}$, where each $z \in \mathcal{Z}$ is the skill embedding of corresponding reference motion in $\mathcal{M}$. We define a unified reorientation $\phi$, to indicate the different stages of one motion, where each stage is defined as every 30 frames. During training, we make it mandatory for the controller to render appropriate actions to make the generated motion sequence $(s, s')$ match both of the specified $z$ and the motion phase $\phi$ with the corresponding motion clip $m$. A discriminator $D((s, s'), z_t, \phi)$ is built, using conditional adversarial motion priors, as the reward generator to verify control policy's ability and guide control policy training. The loss function of the discriminator can be defined as :

$$
\begin{aligned}
\mathcal{L}_D = \mathbb{E}_{p_{\mathcal{M}}(\mathbf{m})} \Big[ &- \mathbb{E}_{p_{\mathrm{m}}(\mathbf{s}, \mathbf{s}', \phi)} \left[ \log \left( D\left( \mathbf{s}, \mathbf{s}', \mathbf{z}, \phi \right) \right) \right] \\
&- w_D \mathbb{E}_{p_{\pi}(\mathbf{s}, \mathbf{s}' | \mathbf{z}, \phi)} \left[ \log \left( 1 - D\left( \mathbf{s}, \mathbf{s}', \mathbf{z}, \phi \right) \right) \right] \\
&- (1 - w_D) \, \mathbb{E}_{p_m(\mathbf{s}, \mathbf{s}', \omega(m), \omega(\phi))} \left[ \log \left( 1 - D\left( \mathbf{s}, \mathbf{s}', \mathbf{z}, \phi \right) \right) \right] \\
&+ w_{\mathrm{gp}} \mathbb{E}_{p_{\mathrm{m}}(\mathbf{s}, \mathbf{s}', \phi)} \left[ \left\| \nabla_{\psi} D(\psi, \mathbf{z}) |_{\psi = (\mathbf{s}, \mathbf{s}', \phi)} \right\|^2 \right] \Big]
\end{aligned}
\tag{1}
$$

It emphasizes the discriminator to recognize the input $(\mathbf{s}, \mathbf{s}', \mathbf{z}, \phi)$ whether $(\mathbf{s}, ss')$, $\mathbf{z}$ and $\phi$ is matched with each other, and whether the collated data is from reference dataset $\mathcal{M}$ or is from the physically simulated humanoid character. The $p_{\mathrm{m}}(\mathbf{s}, \mathbf{s}', \phi)$ indicates the probability of a states transition of a selected reference motion $m$ with its embedding $z$ during stage $\phi$, and $p_{\pi}(\mathbf{s}, \mathbf{s}' \mid \mathbf{z}, \phi)$ denotes the probability of a state transition from the policy $\pi$ with the selected motion embedding $z$ and the motion phase $\phi$. We donate $\omega(x)$ is a function to randomly re-sample x, then $p_m(\mathbf{s}, \mathbf{s}', \omega(m), \omega(\phi))$ represents the probability of a situation that for a states transition from a motion $\omega(m)$ and a motion phase $\omega(\phi)$, any one of them fails to match with the others. The final term of the loss function is a gradient penalty, applied to stabilize the adversarial training. The discriminator is trained along with the control policy, generating the implicit life-like motor skills learning reward:

$$
r_t = -\log \left( 1 - D(\mathbf{s}_t, \mathbf{s}_{t+1}, \mathbf{z}, \phi]) \right).
\tag{2}
$$

Both policy and discriminator are expected to not only distinguish the current phase but also comprehend the temporal relationships between different phases, we use positional-encoding fashion representation to encode motion phase information. Furthermore, we add a special phase, denoted as $\phi^*$, which indicates no specific phase information. The purpose of this design is to retain the policy's original acute adaptive ability to make the most comfortable action that facilitates smooth skill switching, as shown in PADL.

The design of incorporating skill embedding with the motion phase into AMP makes it possible for a more precise motor skill execution. By specifying the target skill embedding with the periodic increasing motion phase $\phi$, the controller is able to manipulate the character to completely execute the target skill and determine how many times to perform this in a physically simulated environment. All of these critical attributes ensure a perfect low execution when integrating the controller into our framework. We further verify these benefits and compare our approach to other low-level controllers in subsection 4.3.

### 3.2 Introspective reasoner

The introspective reasoner consists of core components 3.2.1: planner, picker, and programmer. It internally integrates an advanced trial-and-error-based self-correction mechanism 3.2.3 to align the world knowledge inside LLMs with physics-based tasks, as shown in Figure 3.

Since LLMs can only read and output text information, we have defined skill APIs and sensor APIs as interfaces for the introspective reasoner to call the low-level controller, with each item structured in the form of `<name><description>`. Based on world knowledge and reasoning from the task and environmental feedback in text form, the introspective reasoner selects the appropriate APIs to interact with the low-level controller and the environment.

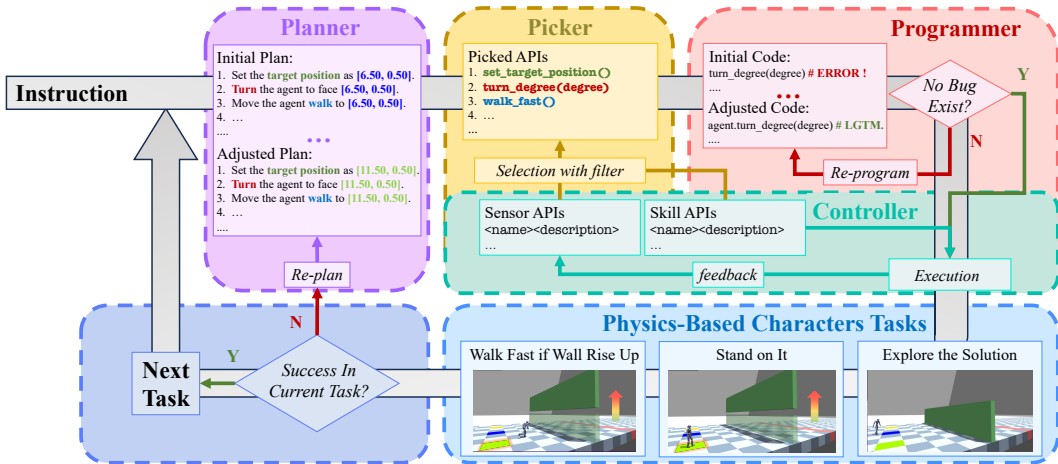

Figure 3: The humanoid character, comprising a planner, picker, programmer, and low-level controller, tries to explore the appropriate pressure plates to raise the walls using sensor APIs and skill APIs, and then pass through. **[6.50, 0.50]** represents an incorrect pressure plates, while **[11.50, 0.50]** denotes the appropriate one.

### 3.2.1 CORE COMPONENTS

**Planner** decomposes the initial instruction into a series of sub-goals based on all the skills the humanoid character has mastered. This process involves high-level task decomposition, enabling LLMs to gain a preliminary understanding of the task flow.

**Picker** sifts through the skill APIs and sensor APIs to select the necessary skills required to achieve those sub-goals, relying on the information provided by the planner for each sub-goal. Considering LLMs' hallucinations, we filter out items in the selected APIs not defined in sensor APIs and skill APIs.

**Programmer** reads the textual descriptions of the sub-goals provided by the planner and the APIs selected by the picker to generate an integrated code for the overall task. This code represents a combination of skills and actions required to complete the plan.

After humans provide the initial instructions for a given task, the planner generates a plan for completing this instruction. For each stage in the plan, the picker selects the appropriate APIs to execute. The programmer then generates the code to execute the plan based on the selected skills. If achieving the initial instruction requires performing other tasks, the introspective reasoner calls the above process multiple times, gradually achieving the initial instruction.

### 3.2.2 SKILL ADJUSTMENT

Given LLMs' limited access to observational data regarding the humanoid character's end effector, the accuracy of skill calls for interacting with objects is compromised.

Consequently, we utilize the introspective reasoner to invoke the low-level controller, adjusting skill APIs based on feedback from sensor APIs, including angles and distances. This process ensures precise execution of low-level actions while showcasing the introspective reasoner's capacity for aligning world knowledge with the physics-based environment. An illustrative example is shown in Figure 4.

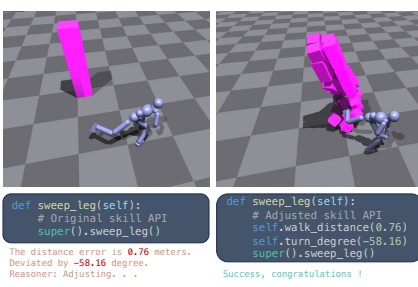

Figure 4: `sweep_leg()` adjustment.

### 3.2.3 TRIAL-AND-ERROR-BASED SELF-CORRECTION

Since LLMs can only access textual descriptions, which provide an incomplete representation of the environment. As a result, the plan and the code generated may not work in one shot.

The trial-and-error-based self-correction, composed of *re-plan* and *re-program* mechanism, automatically achieves alignment between LLMs' world knowledge and physics-based tasks. If the code execution fails, the programmer will *re-program* based on error information. If the current step fails, the planner will *re-plan*s based on environmental feedback. After several rounds of trial-and-error corrections, the introspective reasoner can align itself with the physics-based environment, mitigating the effects of hallucinations caused by LLMs.

## 4 EXPERIMENTS

In this section, we first walk through the quality of our enhanced low-level controller, then we evaluate the effectiveness of our whole system on several real-world tasks. Finally, we highlight the significance of the unified motion phase information for the controller, and why skill adjustment becomes an indispensable component of interactive skills.

### 4.1 LOW-LEVEL CONTROLLER

Our primary focus of assessment lies in evaluating three distinct capabilities of the controller: 1)its proficiency in mastering a multitude of semantically distinguishable skills. 2)its capacity to execute prescribed actions as directed. 3)its ability to swiftly and precisely respond to signals for skill switching, culminating in the flawless and accurate execution of the next skill.

**Dataset**: We train our enhanced low-level controller on a processed MoCap dataset, where each motion clip represents one distinct skill with an average motion length of about 4.2 seconds. The dataset covers a wide range of human behaviors and skills across various categories, from locomotion skills like run, jump, crawl; to dance movements like jazz, hip-pop dance, ballet; as well as some sports like breaststroke, pistol squat, weightlifting, warm-up; as well as many acknowledged challenging or acrobatic skills like jump turn 360 degree, jump spin kick, hook kick, etc, totally of 124 motion clips.

**Our controller learns over 100 various skills in high quality**: Our results reveal that our controller exhibits the capacity to master an extensive repertoire of over 100 semantically distinct motor skills in high quality, exquisitely replicating the authenticity and the completeness of the reference motions. Upon receiving instructions for the execution of specific skills, the controller demonstrates a remarkable ability to swiftly adjust its posture, seamlessly switching to the designated skill in a smooth transition. Furthermore, through the incorporation of motion phase information, the controller achieves a high degree of accuracy in performing the next skill from the correct beginning and has the potential to ensure the completeness of all skills.

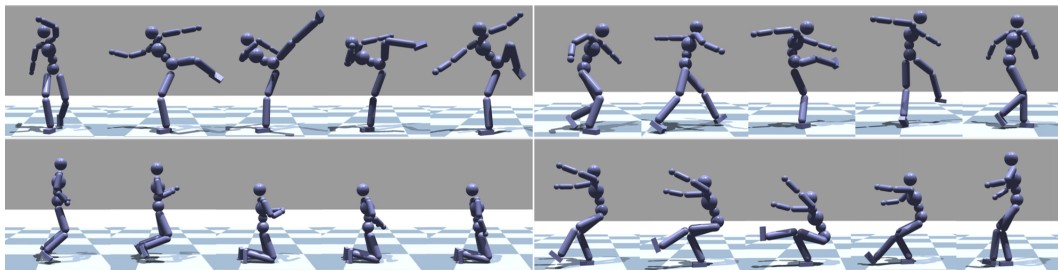

Figure 5: Simulated characters demonstrating various skills from top to bottom: left high side kick, penalty kick, kneel down, pistol squat.

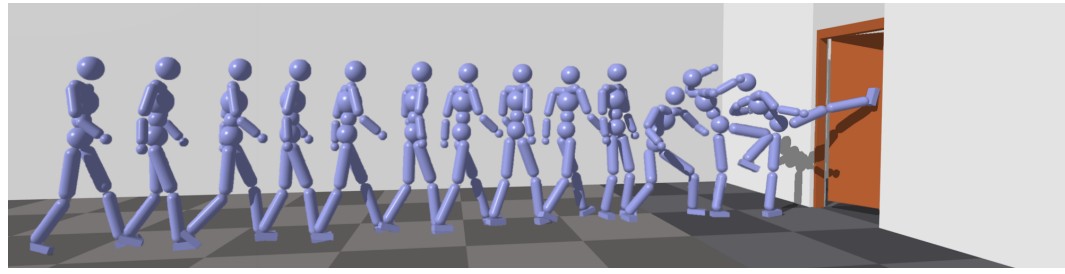

Figure 6: The simulated character approaches the door, uses the "kick forward" skill, and kicks the door open.

## 4.2 REASONING, BEHAVING WITH INTROSPECTION

A successful framework not only enables the deductive reasoning of logical pathways to accomplish tasks but also adeptly orchestrates the utilization of the available motor skill library and sensor tools to fulfill its objectives. Furthermore, The framework should possess the capability for in-context learning, leveraging feedback and historical interactions to deduce novel execution strategies and promptly adapt its tactics to resolve the assigned tasks.

**Tasks design**: We craft a series of tasks aiming to evaluate the task-solving ability of our framework, from rudimentary task to reasoning-required task: **1)** ***Navigation***, the physically simulated character moves itself to the target position in the map. **2)** ***Knock over***, requiring the character to knock down different columns that appear on the ground. **3)** ***Get off the room***, the character is stuck in a room, he must navigate to the front of the door, and then make the door open to get out. **4)***Pressure plate puzzle*, a huge wall blocks the way, the character needs to find the correct plate and then trigger the wall to lift up and move through this obstacle. **5)** ***Knock that over***, there are many columns in the map with different colors, heights, or shapes. The character should follow the instructions and figure out which column should be knocked over. **6)** ***Warrior***, where the character is in a long, narrow apartment filled with various obstacles, requiring the character to escape through all the barriers.

**Our approach helps character solve various tasks in human-like behavior**: For all tasks, we use GPT-4 as our LLMs-based reasoner and adopt the pre-trained low-level controller, with conditional adversarial motion priors, as the control policy for humanoid character. For rudimentary tasks, such as navigation, knock over and get off the room, our framework generally accomplishes correct logical action sequences in a zero-shot manner. However, it is still susceptible to potential flaws stemming from the low-level controller, such as executing "walk" skill will gradually deviate to the left, potentially resulting in mission failure. In the subsequent reasoning, our methodology introspects itself, leveraging environmental feedback and historical interactions, adding a patch of "immediate facing direction checking" into the previous code thought, automatically compensating for the shortcomings of low-level control. For reasoning required tasks, such as pressure plate puzzle and knock over, our system successfully discerns the underlying pattern and finishes the tasks. An interesting finding during the execution of the pressure plate puzzle is that occasionally, the character initially explores the plate that can trigger the elevation of the wall. However, due to insufficient time, the wall descends once more. In response, the character proceeds directly toward the identified correct plate without reiterating the exploration process, then uses running skills to accomplish the task.

Table 1: Pose error comparison in meters.

| Motion | $T_{cycle}(s)$ | Ours |
|---|---|---|
| Dance | 6.10 | 0.071 |
| Run | 1.30 | 0.076 |
| Walk | 1.33 | 0.062 |
| Walk-Fast | 1.93 | 0.045 |
| Throw | 3.90 | 0.038 |
| Walk-then-Jump | 4.70 | 0.075 |
| Jump-Spin-Kick | 2.66 | 0.064 |
| Spin-Hook-Kick | 2.03 | 0.049 |
| Left-Side-Kick | 1.60 | 0.055 |

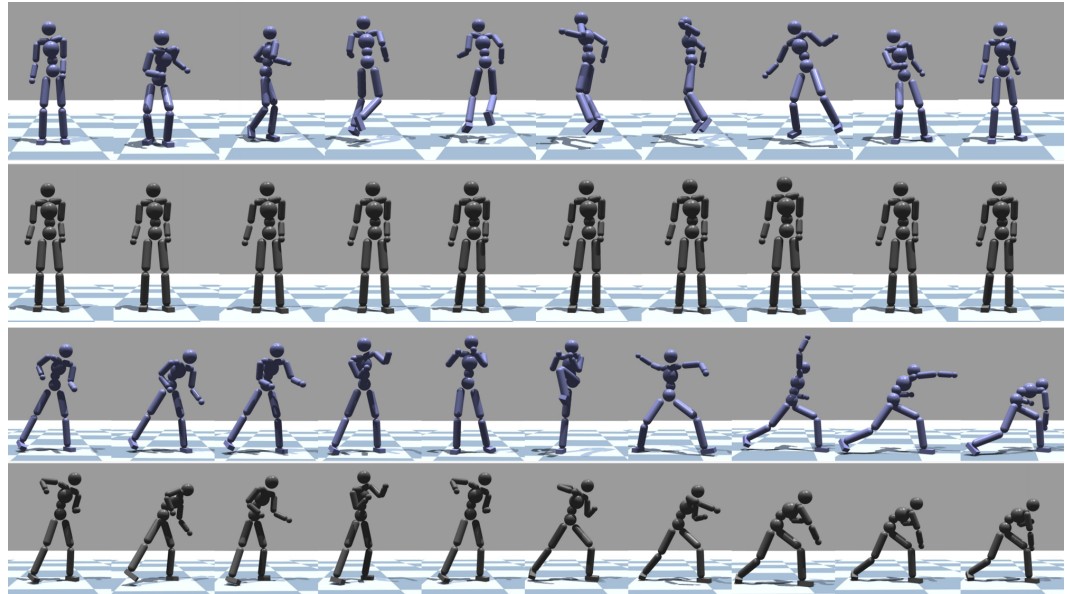

Figure 7: The comparsion of the controllers that miss this crucial information, our control policy is capable of performing a complete skill, rather than being stuck in a static pose or a loop subsequence motion of the skill.

### 4.3    ABLATION ON UNIFIED MOTION PHASE

Control policies trained using AMP suffer mode collapse, leading to the learning of meaningless static poses or repetitive motion to appease the discriminator at a minimal cost. To address this, some works enforce the states transition $(\mathbf{s}, \mathbf{s}')$ generated by the policy corresponding to specified skill embedding $\mathbf{z}$. However, for longer reference motion sequences, exceeding 2 seconds in length, the mode collapse can still persist. Our approach introduces the motion phase information as an additional supervision. It facilitates the the controller not only to push each state transition distribution to match corresponding to skill embedding but also to memorize the disparate phase feature of different reference motions. Compared to the controllers that miss this crucial information, our control policy is capable of performing a complete skill, rather than being stuck in a static pose or a loop subsequence motion of the skill.

## 5    CONCLUSION

In this study, we construct a framework that consists of a introspective reasoner with a motion phase enhanced controller. Our framework leverages the rich world knowledge of a large language model as a prior, help humanoid to better understand the environment and its own capabilities, enabling it to accomplish a wide array of tasks. To ensure a prefect low-execution, we design a motion phase enhanced low-level controller that can specify which skill to execute determine the start time and repetition count for execution. We hope our work can facilitate to the transfer of powerful reasoning models like LLMs into the humanoid character in physical simulation environments.

### AUTHOR CONTRIBUTIONS

If you'd like to, you may include a section for author contributions as is done in many journals. This is optional and at the discretion of the authors.

### ACKNOWLEDGMENTS

Use unnumbered third-level headings for the acknowledgments. All acknowledgments, including those to funding agencies, go at the end of the paper.

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

## A  APPENDIX

### A.1  PHYSICS-BASED CHARACTER CONTROL

**Background**: Control of physically simulated humanoid characters in non-differentiable virtual physical environments is challenging. Reinforcement learning plays a pivotal role in enabling these characters to acquire human-like skills by iteratively refining control policies through trial and error, such as mimicking reference human motions by imitation learning. Similar to mainstream approaches, our learning process for controlling strategies is defined as a Markov Decision Process (MDP) $\mathcal{M} = \langle \mathcal{S}, \mathcal{A}, \mathcal{P}, \mathcal{R}, \gamma \rangle$, comprising states, actions, transition probabilities, rewards, and a discount factor gamma. At each timestep, the control policy $\pi(a_t|s_t, z_t, \phi)$, parameterized by $\theta$, calculates an action $a_t$ based on current state $s_t$ with other conditions we will explain in section llc, and applies $a_t$ in the character. The black-box simulated environment returns the next state $s_{t+1}$ according to transition probabilities $\mathcal{P}$ with a gained reward $r$. The policy incrementally refines its ability to an impressive controller by maximizing the expected discounted return, where

$p_\theta(\tau) = p(s_0) \prod_{t=0}^{T-1} p(s_{t+1}|s_t, a_t) \pi_\theta(a_t|s_t, z_t, \phi)$ is probability distribution over the set of all possible trajectories.

$$J(\theta) = \mathbb{E}_{\tau \sim p_\theta(\tau)} \left[ \sum_{t=0}^{T} \gamma^t r_t \right],$$

**Action and State**: The humanoid we used comprises 15 rigid links with articulated joints interconnecting them, which collectively construct the entire body. Our control policy $\pi(a_t|s_t, z_t, \phi)$ is a proportional-derivative controller, where the output $a_t$ represents the PD target on degree for all revolute joints. The available state of the character consists of the rotation features and relative position to its root node, all recorded in the local coordinate system. The state also includes the linear and angular velocities of every rigid link. All rotation features are represented in a 6D vector. At each timestep, the $\pi(a_t|s_t, z_t, \phi)$ generates $a_t$, then interact with the environment and get the next state $s_{t+1}$. The transition between states is denoted as $(s, s')$, which also indicates a character's motion during this transition.

## A.2 SKILL LIST

| Skill IDs | Skill Name | Skill IDs | Skill Name | Skill IDs | Skill Name | Skill IDs | Skill Name | Skill IDs | Skill Name | Skill IDs | Skill Name |
|---|---|---|---|---|---|---|---|---|---|---|---|
| 0 | Standing on a leg | 21 | Goalie Throw | 41 | Swing Spin Back Kick | 65 | Joyful Jump | 86 | walk calling |
| 1 | Butterfly Stroke | 22 | Play Golf | 42 | Shuttle Run | 66 | Jump High | 88 | Wave Hip-Hop Dance |
| 2 | Shovel | 23 | Elbow Strike | 43 | Cheer | 67 | High Jump Kick | 89 | Zombie Walk |
| 3 | Basketball dribbling | 24 | Continous Jump | 44 | Chicken Dance | 68 | Macarena Dance | 90 | Warm up Jump |
| 4 | Free Stroke | 25 | Rope Skipping | 45 | Straight Punch | 69 | Left High Side Kick | 91 | Warm up Hand to Foot |
| 5 | Walk(slow) | 26 | Turn Left | 46 | Crouched Sneaking | 70 | Left Side Kick(1) | 92 | Warm up Shoulder |
| 6 | Happy Walk | 27 | Stand | 47 | Dance(2) | 71 | Spin Hook Kick | 95 | Air Squat |
| 7 | Push-up | 28 | Run | 48 | Dance(3) | 72 | Move Greet | 99 | Run Tired |
| 9 | Floor Sweep Kick | 29 | Walk(fast) | 50 | Dance(4) | 74 | Spin Around | 100 | Elbow Attack then Uppercut |
| 10 | Crawl(slow) | 30 | Turn Right | 52 | Dodge Walk | 75 | Pointing | 102 | Left Twice Kick |
| 11 | MMA Kick | 31 | Kick Forward | 54 | Penalty Kick | 76 | Pray | 103 | Kick Soccer Ball |
| 12 | Hold Bar Running | 32 | Walk Backward(fast) | 55 | Hip-Hop Dance(1) | 77 | Punch | 105 | Jog Strafe Left |
| 13 | Side Skip | 33 | Squat | 56 | Hip-Hop Dance(2) | 78 | Fast Straight Punch | 107 | Lazily Push Kick |
| 14 | Walk(circle) | 34 | Walk Backward(slow) | 57 | Hip-Hop Dance(3) | 79 | Bruce Lee Kick | 110 | MMA Roundhouse Kick |
| 15 | Twirling | 35 | Walking Turn | 59 | liver shot | 80 | Rumba Dancing | 111 | Floor Spin |
| 16 | Dance(1) | 36 | Walk Normal | 60 | Knee Strike | 81 | Carry thing run | 113 | Squat on one leg |
| 17 | Approach Jump | 37 | Walk Tiptoe | 61 | Jump Spin Kick | 82 | Samba Dancing | 114 | Upward Kick |
| 18 | Throw | 38 | Spin Jump 180 | 62 | Forward Punch | 83 | Silly Dancing | 115 | Putting Down |
| 19 | Climb | 39 | Jump 360 | 63 | Jazz Dancing | 84 | Stabbing | 117 | Crawl(fast) |
| 20 | Squat | 40 | Pick Up | 64 | Jog | 85 | Swing Dancing | 120 | weight lifting |

Table 2: 100 skills out of all the skills that character has mastered.

