# OpenReview forum: "Reason to Behave: Achieving Human-Like Task Execution for Physics-Based Characters"
_ICLR.cc/2024/Conference — ICLR 2024 Conference Withdrawn Submission_

### Official Review · Reviewer_vRe6 · 2023-10-30

**Soundness:** 1 poor
**Presentation:** 1 poor
**Contribution:** 2 fair
**Rating:** 3
**Confidence:** 5

**Summary:**

This paper proposes an LLM-empowered system for solving humanoid tasks. It uses LLM's planning and coding abilities to select pre-learned humanoid behaviors. First, a phase-conditioned motion embedding space is learned to create a skill module that can perform human motion based on action category and API. Then, an LLM is used to write code to elicit the skill from the Humanoid based on the description of the current task. The system is used to solve a number of simple puzzles and room-scale environments.

**Strengths:**

- I find the incorporation of a phase variable a useful addition to precisely control the execution of a low-level skill.
- The qualitative result shows that the humanoid can solve the respective task by generating sequences of human-like behavior.

**Weaknesses:**

- Some important baselines are missing: it appears to me that approaches like MCP [2] can already learn multiple skills in a single network. Since the goal is to switch between skills based on instructions, a simple one-hot vector should suffice as the low-level controller. Namely, a policy can be trained to learn different skills based on a one-hot vector and learn how to switch between distinct skills during training. It seems like that should suffice, instead of a more complicated framework like CALM. CALM has the added use case of training a steering capability.
- Overall, I think the findings in this paper are not super surprising and are more of a system demo than an academic research paper. We already know LLMs can code (to some extent), reason (to some extent), and be applied as a planner (also, to some extent) [3]. Using it to switch between different actions to solve a few pre-defined environments is not particularly impressive. No success rate for the task is reported as well, and only successful demos are shown.
- The paper is poorly written and rushed. The video demo was not present after around 5 days after the paper deadline. A decent amount of wording is unnatural and seems to come from an LLM.
    - "The feasibility of transferring such reasoning to a more concrete 3D physically simulated environment": this is debatable as this paper also uses an abstraction layer for task solving, and using an abstraction layer for solving high-level tasks has been demonstrated on real-world robots, which is a much more challenging task than in simulation.
    - "Commonly, humanoid controllers struggle with accurately carrying out the reasoner's instruction in a non-differentiable physically simulated environment": this is unsubstantiated. Depending on the reasoner's instruction, simulated humanoids have been able to perform a number of actions like Catch and Carry [1].
    - Inappropriate wording:
        - "precise reasoning," "utmost precise control," "perfect low-level execution"
        - "commonsensical," "endowment," "infallibility of low execution," "constricted," "cultivated"
    - Wrong citation style and missing citation link (line 2 of the introduction).
- Equation 1 requires further explanation. What do each term refer to? What are the $W_D$ weights?

[1] Merel, Josh et al. "Catch & Carry: Reusable Neural Controllers for Vision-Guided Whole-Body Tasks." *arXiv: Artificial Intelligence* (2019): n. pag.

[2] Peng, Xue Bin et al. "MCP: Learning Composable Hierarchical Control with Multiplicative Compositional Policies." *ArXiv* abs/1905.09808 (2019): n. pag.

[3] Ahn, Michael et al. "Do As I Can, Not As I Say: Grounding Language in Robotic Affordances." *Conference on Robot Learning* (2022).

**Questions:**

- I suggest reporting the success rate of task solving and expand the range of tasks tested. Also, the task environment should be randomized.
- Improve wording and description.

---

### Official Review · Reviewer_oeTZ · 2023-10-30

**Soundness:** 2 fair
**Presentation:** 1 poor
**Contribution:** 1 poor
**Rating:** 3
**Confidence:** 4

**Summary:**

The paper addresses the challenge of designing physical based character that behave in a human-like manner. The paper uses the LLM as the high-level policy to make use of the low-level control policy. With the LLM reasoner, the character can execute various long horizon tasks specified by language.

**Strengths:**

Using LLM to extend the framework similar to PADL would grant more generality.

**Weaknesses:**

1. This submission appears to be incomplete:
    1. Typos throughout the whole paper, including but not limited to:
        - Misuse of \cite instead of \citep
        - Missing citations in the first paragraph
        - donate ω(x)
        - (s,ss’)
        - “Unified reorientation”

    2. The paper's clarity leaves much to be desired:
        - What exactly is “humanoid character’s end effector”?
        - Table 1 lacks any reference within the text.
        - In the introduction, it is never clear what the reasoner is.
         - The motivation behind the phase representation remains nebulous until Section 4.3. Even within Section 4.3, the justification for this representation is inadequate and lacks crucial validation.
        - It is never clear what the unified phase representation is. Is it a number or a vector?
        - Which dataset are you using?
        - The whole section describing the approach is blurry.

    3. Missing references including but not limited to:
        - All kinds of variants of “code as policy” [1]
        - All kinds of variants of “LLM as task planner” [2]

    4. Missing experiment validation, there is no proper evaluation of the proposed approach.
        - If the paper claims contribution on the low-level controller. Comparison needs to be presented with ASE or AMP.
        - If the paper claims contribution on the high level reasoning. Comparison needs to be presented with all the variants using LLM as task planner.

2. The current presentation of the manuscript obscures its technical merit. However, here are some points that seem related to its main contributions:
    1. The phase-aware policy was previously introduced in the original DeepMimic paper. What exactly differentiates this work, and why is this distinction significant?
    2. Many of the techniques employed for high-level planning in this paper appear similar to existing task planning methodologies using LLM. How does the proposed method uniquely cater to physical character control?

[1] Liang et al., Code as Policies: Language Model Programs for Embodied Control, 2023.

[2] Ahn et al., Do As I Can, Not As I Say: Grounding Language in Robotic Affordances, 2022.

**Questions:**

See weaknesses.

---

### Official Review · Reviewer_qhTT · 2023-10-30

**Soundness:** 1 poor
**Presentation:** 1 poor
**Contribution:** 1 poor
**Rating:** 1
**Confidence:** 4

**Summary:**

This paper presents a method for combining large language models as general-purpose reasoners with a low-level controller for executing a large variety of different skills. The low-level controller is trained with a conditional adversarial objective, allowing it to be robustly used for simultaneously representing many different skills from a motion dataset successfully. The large language model translates given instructions into a plan consisting of multiple steps, which a “picker” then translates to a sequence of API calls based on pre-defined sensor and skill APIs, and this is finally translated into an executable program consisting of a sequence of actions and skills to be applied. If the first application fails, trial-and-error is used to update the plan and the program to move the system closer to a solution. The approach is demonstrated on a collection of humanoid navigation tasks.

**Strengths:**

The paper is tackling an interesting and significant problem – how to perform long-horizon tasks in robotics. While the idea of using skills is not novel, it is nice to see skills being combined with a large language model reasoner to try to expand the scope of what robots can do in simulation. Many people in the robotics community are currently interested in these questions, so the paper would have a large audience based on the topic alone. The website has very nice videos that are fun to watch.

**Weaknesses:**

While the topic is of great interest, this paper is incomplete, and unclear to the point of being uninterpretable and unreproducible. It appears that the paper was very rushed for submission, and there are many basic formatting / grammar issues making the paper hard to read. There are also significant undefined parts of the method, and no quantitative results or comparisons to alternative approaches. I give a few examples of each issue below. I think the overall approach has promise, and I would encourage the authors to spend significant time in editing the paper before resubmitting in the future.

Basic formatting and grammar issues
* All citations are incorrectly formatted, without parentheses around them in the text. This makes the text hard to follow and hard to interpret consistently, throughout the paper.
* Author contributions and acknowledgements are left in the paper with boilerplate text included.
* Several missing citations, showing up as ?s in the document
* Unfinished sentences. E.g. “In this section, we will elaborate on modules in our framework according to the following workflow. We first inspect the design details of skill embedding and unified motion phase representation conditioned low-level controller in subsection 3.1, focusing on its .”
* Several presumably wrong uses of words, such as “fashion representation”, which I have never heard of before

Significant undefined parts of the method

Adversarial motion prior component
* The paper describes the conditional adversarial motion prior reasonably precisely, although with key pieces of missing information. For example, the paper mentions using MotionCLIP, but this is both undefined and uncited. Similarly, I did not understand some of the definitions (e.g. “We define a unified reorientation ϕ, to indicate the different stages of one motion, where each stage is defined as every 30 frames” – what is a reorientation?)
* inconsistent notation with whether there are time subscripts made it difficult to determine which equations were over sets or individual timepoints.
* “We donate ω(x) is a function to randomly re-sample x, then pm(s, s ′ , ω(m), ω(ϕ) represents the probability of a situation that for a states transition from a motion ω(m) and a motion phase ω(ϕ),” – what is the resampling function? I did not understand what this meant.
* “By specifying the target skill embedding with the periodic increasing motion phase ϕ” – now phi is a motion phase instead of unified reorientation, and it is constrained to be periodic?

Planner, Picker, Programmer, Skill adjustment and trial-and-error learning
* Almost no description of how Planner, Picker, Programmer work at all.
  * “Planner decomposes the initial instruction into a series of sub-goals based on all the skills the humanoid character has mastered. This process involves high-level task decomposition, enabling LLMs to gain a preliminary understanding of the task flow. Picker sifts through the skill APIs and sensor APIs to select the necessary skills required to achieve those sub-goals, relying on the information provided by the planner for each sub-goal.”
  * This is simply not enough information for a reader to determine how the LLM was used to do the decomposition and the picking of the APIs. This needs to be significantly expanded.
* Skill adjustment.
  * Paper claims that skill adjustment is done by receiving feedback about errors from the task execution. However, there is no further information about how this is formatted for the LLM, or even which part of the planner, picker, programmer pipeline it is going to. I assume that it’s just a formulaic text description, but it is hard to tell.
* Trial-and-error.
  * “The trial-and-error-based self-correction, composed of re-plan and re-program mechanism, automatically achieves alignment between LLMs’ world knowledge and physics-based tasks. If the code execution fails, the programmer will re-program based on error information. If the current step fails, the planner will re-plans based on environmental feedback. After several rounds of trial-and-error corrections, the introspective reasoner can align itself with the physics-based environment, mitigating the effects of hallucinations caused by LLMs.”
  * This is not enough detail for a reader. How is the error formatted? When is there re-planning vs. re-programming?

No explained quantitative results, no comparisons to alternative approaches
* While the paper provides a qualitative analysis of some of the tasks, there are no quantitative results that are discussed in the text. There is a table, Table 1, with tasks, an undefined column “T_cycle” and “ours”, but this is never referred to in the text, and it is not clear what the numbers in the table mean, or how they should be interpreted. There is no additional information to this effect in the appendix.
* While the paper mentions several closely related works, none of them are compared to in the experiments or results, and none of the tasks studied are standard tasks for humanoid robots (which would ease comparison to other papers)

**Questions:**

Questions are included with "weaknesses" above.

---

### Official Review · Reviewer_DdyJ · 2023-11-01

**Soundness:** 3 good
**Presentation:** 3 good
**Contribution:** 2 fair
**Rating:** 5
**Confidence:** 4

**Summary:**

The paper tackles the challenge of achieving human-like task execution in different puzzle scenarios for physics-based characters. The authors propose a pipeline called "Reason to Behave" that integrates a low level controller that can imitate a library of behaviors on demand with LLMs to leverage text description of the task, and text description of the task feedback from the environment. Key aspects the authors highlight are the reasoner's capacity for leveraging back prompting to mimic contextual reasoning and self-correction. Experiments demonstrate the framework's effectiveness across tasks requiring reasoning, planning, and natural movement, though some reliance on human oversight, tuning and environment design persists.

**Strengths:**

**Large motor skill repertoire:** The controller can robustly replicate over 100 distinct motor skills, ranging from locomotion to acrobatics. The proposed architecture is a welcome step forward on bridging the gap between imitation objective and skill embedding work. This skill diversity and robustness allows for composing these behaviors for solving versatile downstream tasks.

**Integration of LLMs and controllers:** The proposed pipeline leverages the structure of Latent Space (meaningful z , 1-1 correspondence to skills) for designing code based policies for downstream tasks. They showcase the capacity of LLMs to understand this structure that enables proposal plans and refinement using back prompting environment feedback in the form of text.

**Diverse downstream tasks:** The pipeline is tested on a wide variety of downstream tasks, from basic navigation to relatively complex puzzles to demonstrates generalization of the proposed pipeline. By leveraging LLMs and back prompting, the system demonstrates behaviors similar mimic contextual human reasoning and self-correction.

**Weaknesses:**

- The proposed approach does not compare itself to any baselines such as just using a tracking reward with imitation to be able to imitate these behaviors. What would be the pros and cons of this. Can tracking reward be integrated with Style loss as some other related works. It would be wonderful if authors could provide some discussion about the advantages and disadvantages of the proposed architecture. Can this be able to imitate just one pose from the trajectory and not mimic the entire trajectory?, Could we use this to solve heading task? (move in a certain heading while facing certain direction?)

- The details of how the skill and sensor APIs are created are unclear. The paper states they are predefined, but does not list the APIs. It seems they were manually defined by the authors based on the skill library. How are we created apis that take a parameter such as walk_distance, turn_degree. More implementation details on the APIs would improve reproducibility.

- Are we overselling this? The approach is demonstrated for structured puzzle scenarios that provide textual feedback for the LLM. While the approach does pave a pipeline on using LLMs to generate code and using them as Finite State Machine sub policies to achieve a sub task, The application is still limited by the need of textual feedback mechanisms. While I do understand the notion of authors the word “plan” may be a bit of an oversell for some readers from ICAPS community. It is important to have some discussion of the limitations and assumptions of the current approach.

- It is important to distinguish the novelty of the work separately on the use of phase informed imitation learning and LLM integration as it is fairly established at this time that we can use LLMs to leverage the pre defined robot APIs to come up with complex behaviors. It is interesting to view the second half of the work as a pipeline to generate higher level APIs / Tasks specific APIs that can be further added to the API repertoire, which may be a more general value proposition that is invariant of the current low level controller.

**Questions:**

NA